# Improved CUT&RUN chromatin profiling tools

Michael P Meers[1], Terri D Bryson[1,2], Jorja G Henikoff[1], Steven Henikoff[1,2]*

[1]Basic Sciences Division, Fred Hutchinson Cancer Research Center, Seattle, United States; [2]Howard Hughes Medical Institute, United States

**Abstract** Previously, we described a novel alternative to chromatin immunoprecipitation, CUT&RUN, in which unfixed permeabilized cells are incubated with antibody, followed by binding of a protein A-Micrococcal Nuclease (pA/MNase) fusion protein (Skene and Henikoff, 2017). Here we introduce three enhancements to CUT&RUN: A hybrid protein A-Protein G-MNase construct that expands antibody compatibility and simplifies purification, a modified digestion protocol that inhibits premature release of the nuclease-bound complex, and a calibration strategy based on carry-over of *E. coli* DNA introduced with the fusion protein. These new features, coupled with the previously described low-cost, high efficiency, high reproducibility and high-throughput capability of CUT&RUN make it the method of choice for routine epigenomic profiling.
DOI: https://doi.org/10.7554/eLife.46314.001

## Introduction

Profiling the chromatin landscape for specific components is one of the most widely used methods in biology, and over the past decade, chromatin immunoprecipitation (ChIP) followed by sequencing (ChIP-seq) has become practically synonymous with genome-wide chromatin profiling (*Landt et al., 2012*; *Schubert, 2018*). However, the most widely used ChIP-seq protocols have limitations and are subject to artifacts (*Jain et al., 2015*; *Park et al., 2013*; *Teves et al., 2016*; *Teytelman et al., 2013*), of which only some have been addressed by methodological improvements (*Brind'Amour et al., 2015*; *Kasinathan et al., 2014*; *Rhee and Pugh, 2011*; *Rossi et al., 2018*; *van Galen et al., 2016*). An inherent limitation to ChIP is that solubilization of chromatin, whether by sonication or enzymatic digestion, results in sampling from the entire solubilized genome, and this requires very deep sequencing so that the sites of targeted protein binding can be resolved above background (*Landt et al., 2012*). To overcome this limitation, we introduced Cleavage Under Targets and Release Using Nuclease (CUT&RUN) (*Skene and Henikoff, 2017*), which is based on the chromatin immunocleavage (ChIC)-targeted nuclease strategy (*Schmid et al., 2004*): Successive incubation of unfixed cells or nuclei with an antibody and a Protein A-Micrococcal Nuclease (pA/MNase) fusion protein is followed by activation of MNase with calcium. In CUT&RUN, cells or nuclei remain intact throughout the procedure and only the targeted sites of binding are released into solution. Our CUT&RUN method dramatically reduced non-specific backgrounds, such that ~10 fold lower sequencing depth was required to obtain similar peak-calling performance (*Skene and Henikoff, 2017*). In addition, CUT&RUN provides near base-pair resolution, and our most recently published benchtop protocol is capable of profiling ~100 human cells for an abundant histone modification and ~1000 cells for a transcription factor (*Skene et al., 2018*). The simplicity of CUT&RUN has also resulted in a fully automated robotic version (AutoCUT&RUN) in which the high reproducibility and low cost makes it ideally suited for high-throughput epigenomic profiling of clinical samples (*Janssens et al., 2018*). Other advances based on our original CUT&RUN publication include CUT&RUN.Salt for fractionation of chromatin based on solubility (*Thakur and Henikoff, 2018*) and CUT&RUN.ChIP for profiling specific protein components within complexes released by CUT&RUN

*For correspondence:
steveh@fhcrc.org

Competing interests: The authors declare that no competing interests exist.

digestion (*Brahma and Henikoff, 2019*). CUT&RUN has also been adopted by others (*Ahmad and Spens, 2018*; *Daneshvar et al., 2019*; *de Bock et al., 2018*; *Ernst et al., 2019*; *Federation et al., 2018*; *Hainer et al., 2019*; *Hainer and Fazzio, 2019*; *Hyle et al., 2019*; *Inoue et al., 2018*; *Liu et al., 2018*; *Menon et al., 2019*; *Oomen et al., 2019*; *Park et al., 2019*; *Roth et al., 2018*; *Uyehara and McKay, 2019*; *Zhang et al., 2019*; *Zheng and Gehring, 2019*), and since publication of our *eLife* paper we have distributed materials to >600 laboratories world-wide, with user questions and answers fielded interactively on our open-access Protocols.io site (dx.doi.org/10.17504/protocols.io.zcpf2vn).

Broad implementation of CUT&RUN requires reagent standardization, and the rapid adoption of CUT&RUN by the larger community of researchers motivates the enhancements described here. First, the method requires a fusion protein that is not at this writing commercially available, and the published pA/MNase purification protocol is cumbersome, which effectively restricts dissemination of the method. Therefore, we have produced an improved construct with a 6-His-Tag that can be easily purified using a commercial kit, and by using a Protein A-Protein G hybrid, the fusion protein binds avidly to mouse antibodies, which bind only weakly to Protein A. Second, the original protocols are sensitive to digestion time, in that under-digestion results in low yield and over-digestion can result in pre-mature release of pA/MNase-bound complexes that can digest accessible DNA sites. To address this limitation, we have modified the protocol such that premature release is reduced, allowing digestion to near-completion for high yields with less background. Third, the current CUT&RUN protocol recommends a spike-in of heterologous DNA at the release step to compare samples in a series. Here we demonstrate that adding a spike-in is unnecessary, because the carry-over of *E. coli* DNA from purification of pA/MNase or pAG/MNase is sufficient to calibrate samples in a series.

## Results and discussion

### An improved CUT&RUN vector

The pA/MNase fusion protein produced by the pK19-pA-MN plasmid (*Schmid et al., 2004*) requires purification from lysates of *Escherichia coli* overexpressing cells using an immunoglobulin G (IgG) column, and elution with low pH followed by neutralization has resulted in variations between batches. To improve the purification protocol, we added a 6-His tag (*Bornhorst and Falke, 2000*) into the pK19-pA-MN fusion protein (*Figure 1A* and *Figure 1—figure supplement 1A*). This allowed for simple and gentle purification on a nickel resin column (*Figure 1—figure supplement 1B*). In addition, we found that a commercial 6-His-cobalt resin kit also yielded pure highly active enzyme from a 20 ml culture, enough for ~10,000 reactions. Even when used in excess, there is no increase in release of background fragments (*Figure 1—figure supplement 2*), which indicates that the washes are effective in removing unbound fusion protein.

In principle an epitope-tagged pAG/MNase could be used for chromatin pull-down from a CUT&RUN supernatant in sequential strategies like CUT&RUN.ChIP (*Brahma and Henikoff, 2019*). However, in practice use of the 6-His tag is complicated by the requirement for a chelating agent to release the protein from the nickel resin. Therefore, we also added an HA (hemagglutinin) tag, which could be used to affinity-purify the complex of a directly bound chromatin particle with a primary antibody and the fusion protein.

Protein A binds well to rabbit, goat, donkey and guinea pig IgG antibodies, but poorly to mouse IgG1, and so for most mouse antibodies, Protein G is generally used (*Fishman and Berg, 2019*). To further improve the versatility of the MNase fusion protein, we encoded a single Protein G domain adjacent to the Protein A domain in the pK19-pA-MN plasmid (*Eliasson et al., 1988*). In addition, we mutated three residues in the Protein G coding sequence to further increase binding for rabbit antibodies (*Jha et al., 2014*). This resulted in a fusion protein that binds strongly to most commercial antibodies without requiring a secondary antibody. We found that for ordinary CUT&RUN applications pAG/MNase behaves very similarly to pA/MNase, but is more easily purified and is more versatile, for example allowing us to perform CUT&RUN without requiring a secondary antibody for mouse primary monoclonal antibodies (*Figure 1B*).

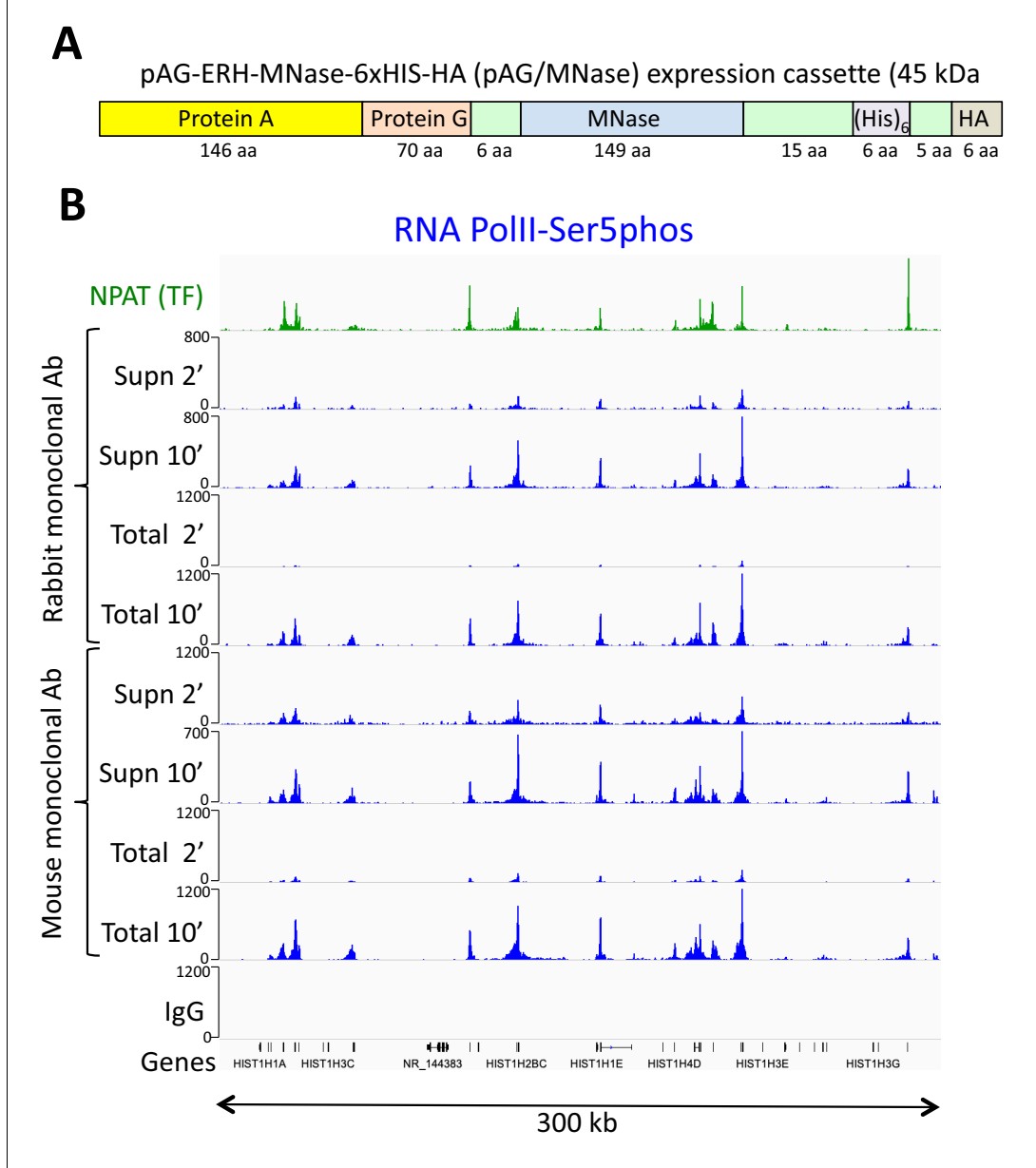

**Figure 1.** An improved fusion protein for CUT&RUN. (**A**) Schematic diagram (not to scale) showing improvements to the pA-MNase fusion protein, which include addition of the C2 Protein G IgG binding domain, a 6-histidine tag for purification and a hemagglutinin tag (HA) for immunoprecipitation. (**B**) The Protein A/G hybrid fusion results in high-efficiency CUT&RUN for both rabbit and mouse primary antibodies. CUT&RUN for both rabbit and mouse RNAPII-Ser5phosphate using pAG/MNase were extracted from either the supernatant or the total cellular extract. Tracks are shown for the histone gene cluster at Chr6:26,000,000–26,300,000, where NPAT is a transcription factor that co-activates histone genes. Tracks for 2' and 10' time points are displayed at the same scale for each antibody and for both supernatant (supn) or total DNA extraction protocols.

DOI: https://doi.org/10.7554/eLife.46314.002

The following figure supplements are available for figure 1:

**Figure supplement 1.** An improved fusion protein for CUT&RUN.
DOI: https://doi.org/10.7554/eLife.46314.003

**Figure supplement 2.** pAG/MNase titration.
DOI: https://doi.org/10.7554/eLife.46314.004

## Preventing premature release during CUT&RUN digestion

When fragments are released by cleavage in the presence of Ca$^{++}$ ions, the associated pA/MNase complex can digest accessible DNA (*Skene and Henikoff, 2017*). Although performing digestion at 0°C minimizes this artifact, eliminating premature release during digestion would allow for more complete release of target-specific fragments. Based on the observation that nucleosome core particles aggregate in high-divalent-cation and low-salt conditions (*de Frutos et al., 2001*), we wondered whether these conditions would prevent premature release of chromatin particles in situ. Therefore, we performed digestions in 10 mM CaCl$_2$ and 3.5 mM HEPES pH 7.5. Under these high-calcium/low-salt conditions, chromatin is digested with no detectable release of fragments into the supernatant (*Figure 2*). Reactions are halted by transferring the tube to a magnet, removing the liquid, and adding elution buffer containing 150 mM NaCl, 20 mM EGTA and 25 μg/ml RNAse A, which releases the small DNA fragments into the supernatant. These conditions are compatible with direct end-polishing and ligation used for AutoCUT&RUN (*Janssens et al., 2018*). Furthermore, retention of the cleaved fragments within the nucleus under high-divalent cation/low-salt conditions could facilitate single-cell application of CUT&RUN.

The high-calcium/low-salt protocol provided similar results using either pA/MNase and pAG/MNase (*Figure 3*). We also obtained similar results with either protocol for digestion time points over a ~ 30 fold range and for both supernatant and total DNA extraction (*Figure 4—figure supplement 1*). For antibodies to H3K27ac, libraries produced using the high-calcium/low-salt protocol showed improved consistency relative to the standard protocol when digested over an extended time-course (*Figure 4*), presumably because preventing release of particles during digestion avoids their premature release where they would artifactually digest accessible DNA. The close correlations between high-calcium/low-salt H3K27ac datasets for time points over a ~ 100 fold range occur with corresponding increases in the yield of fragments released into the supernatant during subsequent elution (*Figure 4—figure supplement 2*). This indicates that longer digestion times result in higher yields, with high signal-to-noise throughout the digestion series (*Figure 4*). Thus, this modification of CUT&RUN can reduce the risk of overdigestion for abundant epitopes such as H3K27ac, where premature release of pA-MNase-bound chromatin particles can increase background.

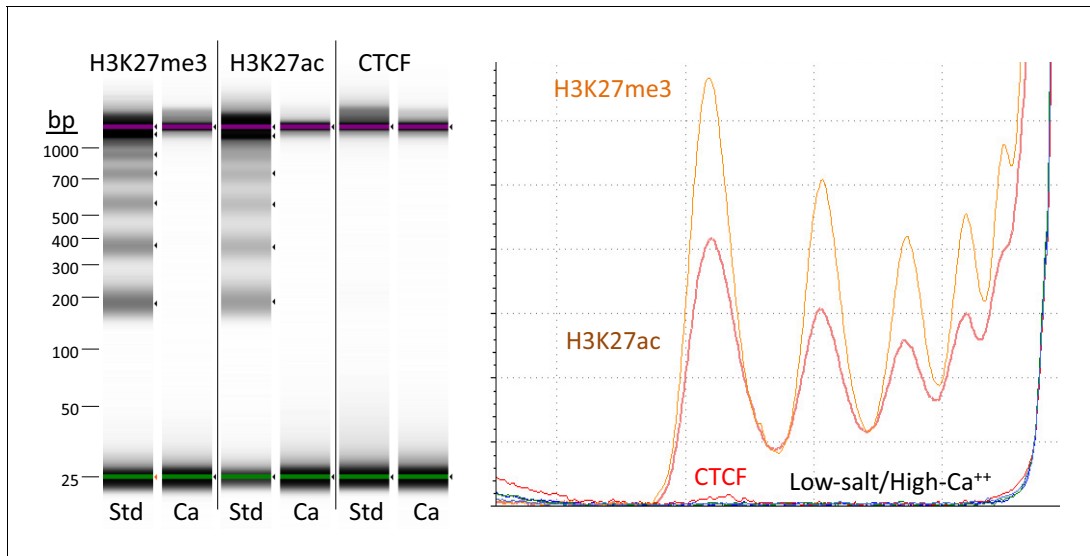

**Figure 2.** Targeted fragments are not released during digestion using high-calcium/low-salt conditions. CUT&RUN was performed using either the high-Ca$^{++}$/low-salt (Ca$^{++}$) or the standard (Std) method with antibodies to three different epitopes. DNA was extracted from supernatants, where no elution was carried out for the Ca$^{++}$ samples. Although high yields of nucleosomal ladder DNA eluted from the supernatants using the standard method, no DNA was detectable in the supernatant using the high-Ca$^{++}$/low salt method when the elution step was omitted. Left, Tapestation images from indicated lanes; Right, Densitometry of the same lanes.
DOI: https://doi.org/10.7554/eLife.46314.005

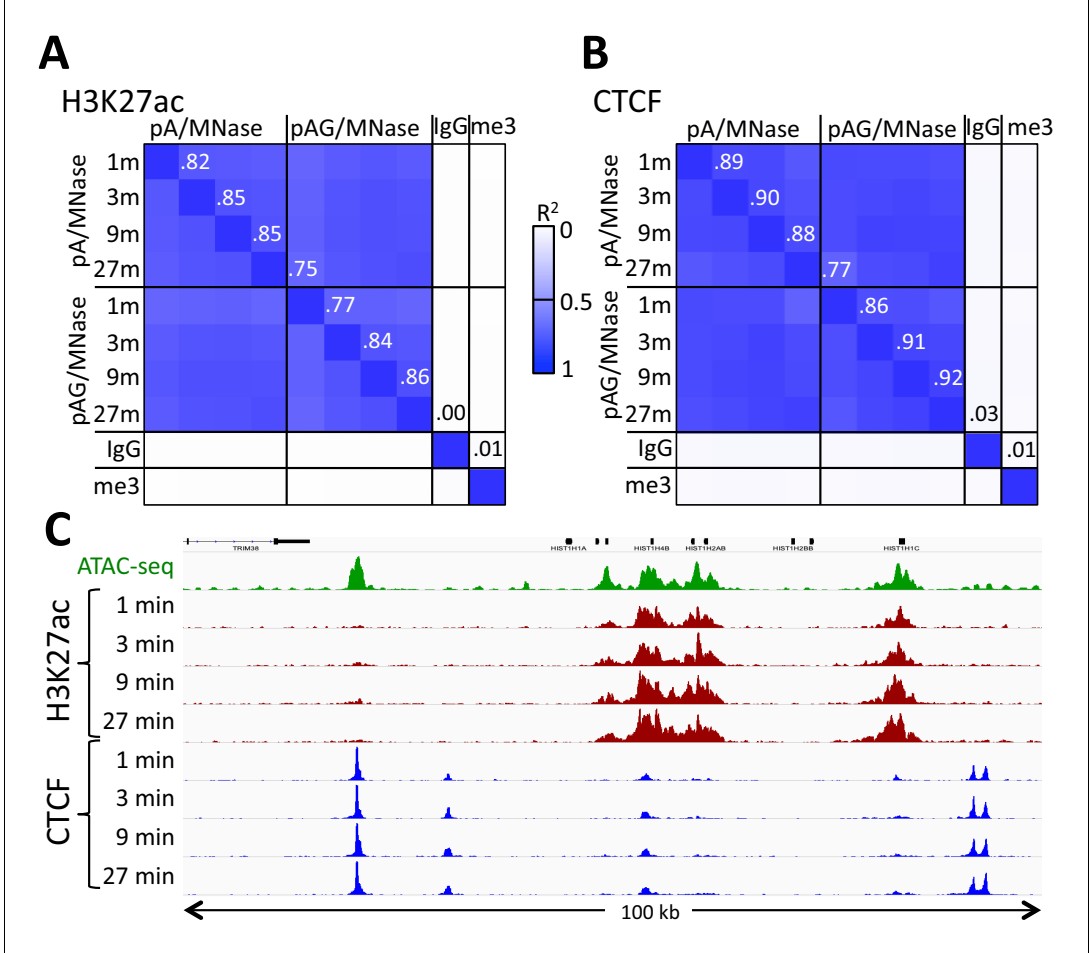

**Figure 3.** Similar performance using pA/MNase and pAG/MNase. (**A**) CUT&RUN was performed with an antibody to H3K27ac (Millipore MABE647) and to CTCF (Millipore 07–729) with digestion over a 1 to 27 min range as indicated using pAG/MNase with the high-Ca⁺⁺/low-salt protocol. Correlation matrix comparing peak overlaps for time points and fusion constructs. The datasets were pooled and MACS2 was used with default parameters to call peaks, excluding those in repeat-masked intervals and those where peaks overlapped with the top 1% of IgG occupancies, for a total of 52,425 peaks. Peak positions were scored for each dataset and correlations ($R^2$ values shown along the diagonal and displayed with Java TreeView v.1.16r2, contrast = 1.25) were calculated between peak vectors. IgG and H3K27me3 (me3) negative controls were similarly scored. (**B**) Same as A, except the antibody was to CTCF. A set of 9403 sites with a CTCF motif within a hypersensitive site was used (*Skene and Henikoff, 2017*). High correlations between all time points demonstrate the uniformity of digestion over a 27-fold range. (**C**) Representative tracks from datasets used for panels A and B showing a 100 kb region that includes a histone locus cluster (chr6:25,972,600–26,072,600).

DOI: https://doi.org/10.7554/eLife.46314.006

We previously showed that CUT&RUN can be performed on insoluble protein complexes by extracting total DNA (*Skene and Henikoff, 2017*) or by performing salt fractionation of the bead-bound cells and extracting DNA from the residual pellet (*Thakur and Henikoff, 2018*). In either case, large DNA fragments were depleted using SPRI (AMPure XP) beads before library preparation. RNA polymerase II (RNAPII) from animal cells is insoluble when engaged (*Mayer et al., 2015*; *Weber et al., 2014*), and requires harsh treatments for quantitative profiling using ChIP (*Skene and Henikoff, 2015*). To determine whether CUT&RUN can be used for insoluble chromatin complexes, we profiled Serine-5-phosphate on the C-terminal domain (CTD) of the Rpb1 subunit of RNAPII using both extraction of supernatant and of total DNA. This CTD phosphorylation is enriched in the initiating form of RNAPII, and we observed similar genic profiles for supernatant and total DNA (*Figure 1B*).

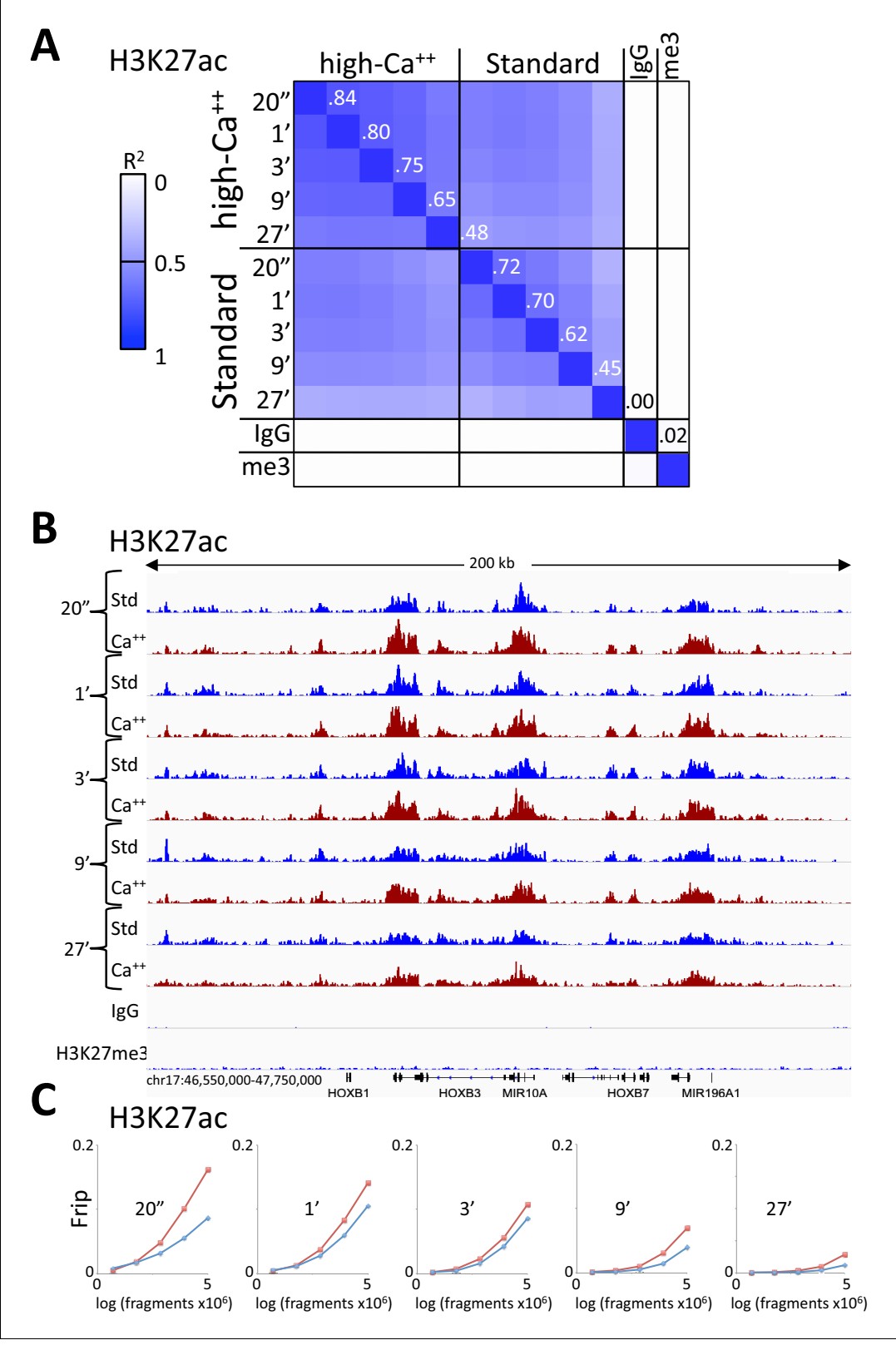

**Figure 4.** Consistent peak definition with high-Ca++/low salt digestion. (**A**) H3K27ac CUT&RUN time-course experiments were performed with an Abcam 4729 rabbit polyclonal antibody, following either the standard protocol or the low-salt/high-calcium (High-Ca++) protocol. Samples of 5 million fragments from the 10 H3K27ac datasets were pooled and MACS2 called 36,529 peaks. Peak positions were scored for each dataset and

*Figure 4 continued on next page*

*Figure 4 continued*

correlations ($R^2$ values shown along the diagonal) were calculated between peak vectors. IgG and H3K27me3 (me3) negative controls were similarly scored. Higher correlations between the High-Ca$^{++}$ than the Standard time points indicates improved uniformity of digestion over the ~100 fold range of digestion times. (B) Tracks from a representative 200 kb region around the HoxB locus. (C) Fraction of reads in peaks (Frip) plots for each time point after down-sampling (5 million, 2.5 million, 1.25 million, 625,000 and 312,500), showing consistently higher Frip values for Ca$^{++}$ (red) than Std (blue).

DOI: https://doi.org/10.7554/eLife.46314.007

The following figure supplements are available for figure 4:

**Figure supplement 1.** CUT&RUN consistency with high-Ca++/low salt digestion and total DNA extraction.
DOI: https://doi.org/10.7554/eLife.46314.008
**Figure supplement 2.** Tapestation analyses of an H3K27ac digestion time-course series.
DOI: https://doi.org/10.7554/eLife.46314.009

## Calibration using *E. coli* carry-over DNA

Comparing samples in a series typically requires calibration for experimental quality and sequencing read depth. It is common to use background levels to calibrate ChIP-seq samples in a series and to define and compare peaks for peak-calling (*Landt et al., 2012*). However, the extremely low backgrounds of CUT&RUN led us to a calibration strategy based on spike-in of heterologous DNA, which has been generally recommended for all situations in which samples in a series are to be compared (*Chen et al., 2015*; *Hu et al., 2015*). In our current spike-in protocol, the heterologous DNA, which is typically DNA purified from an MNase digest of yeast *Saccharomyces cerevisiae* or *Drosophila melanogaster* chromatin, is added when stopping a reaction, and we adopted this spike-in procedure for the high-calcium/low-salt protocol described in the previous section. Interestingly, we noticed that mapping reads to both the spike-in genome and the *E. coli* genome resulted in almost perfect correlation ($R^2$ = 0.97) between *S. cerevisiae* and *E. coli* in an experiment using pA/MNase in which the number of cells was varied over several orders of magnitude (*Figure 5A*). Near-perfect correlations ($R^2$ = 0.96–0.99) between yeast spike-in and carry-over *E. coli* DNA were also seen in series using the same batch of pAG/MNase with high-calcium/low-salt digestion conditions (*Figure 5B*), and for both supernatant release and extraction and total DNA extraction (*Figure 5C–D*). These strong positive correlations are not accounted for by cross-mapping of the yeast spike-in to the *E. coli* genome, because omitting the spike-in for a low-abundance epitope resulted in very few yeast counts with high levels of *E. coli* counts (blue symbol in *Figure 5C–D* panels). As the source of *E. coli* DNA is carried over from purification of pA/MNase and pAG/MNase, the close correspondence provides confirmation of the accuracy of our heterologous spike-in procedure (*Skene and Henikoff, 2017*). Moreover, as carry-over *E. coli* DNA is introduced at an earlier step, and is cleaved to small mappable fragments that are released during digestion and elution, it provides a more desirable calibration standard than using heterologous DNA (*Chen et al., 2015*; *Hu et al., 2015*). High correlations were also seen between *S. cerevisiae* spike-in and *E. coli* carry-over DNA for pA-MNase in batches that we have distributed (*Table 1*). Therefore, data for nearly all CUT&RUN experiments performed thus far can be recalibrated *post-hoc* whether or not a spike-in calibration standard had been added.

To explain the presence of carry-over *E. coli* DNA in proportion to the amount of yeast spike-in DNA, which is constant between samples in a series, we can exclude intracellular binding, because we observe proportionality between *E. coli* and yeast reads despite varying human cell numbers over two orders of magnitude (*Figure 5A*). Rather, we note that Concanavalin A binds to glycosylated immunoglobulins, and so the successive treatments of Con A bead-bound cells with excess antibody and Protein A(G)/MNase fusion protein will affix an amount of carry-over *E. coli* DNA to beads in proportion to the number of beads. Our use of a constant number of beads for all samples in a series to be compared would then have resulted in a constant amount of carry-over *E. coli* DNA. A similar inference of *E. coli* carry-over DNA suitable for calibration was noted for CUT&Tag (*Kaya-Okur et al., 2019*), which suggests successive binding of antibodies and Protein A-Tn5 to the Con A beads used to immobilize cells. Thus, our calibration strategy might serve as a more general replacement for conventional spike-ins.

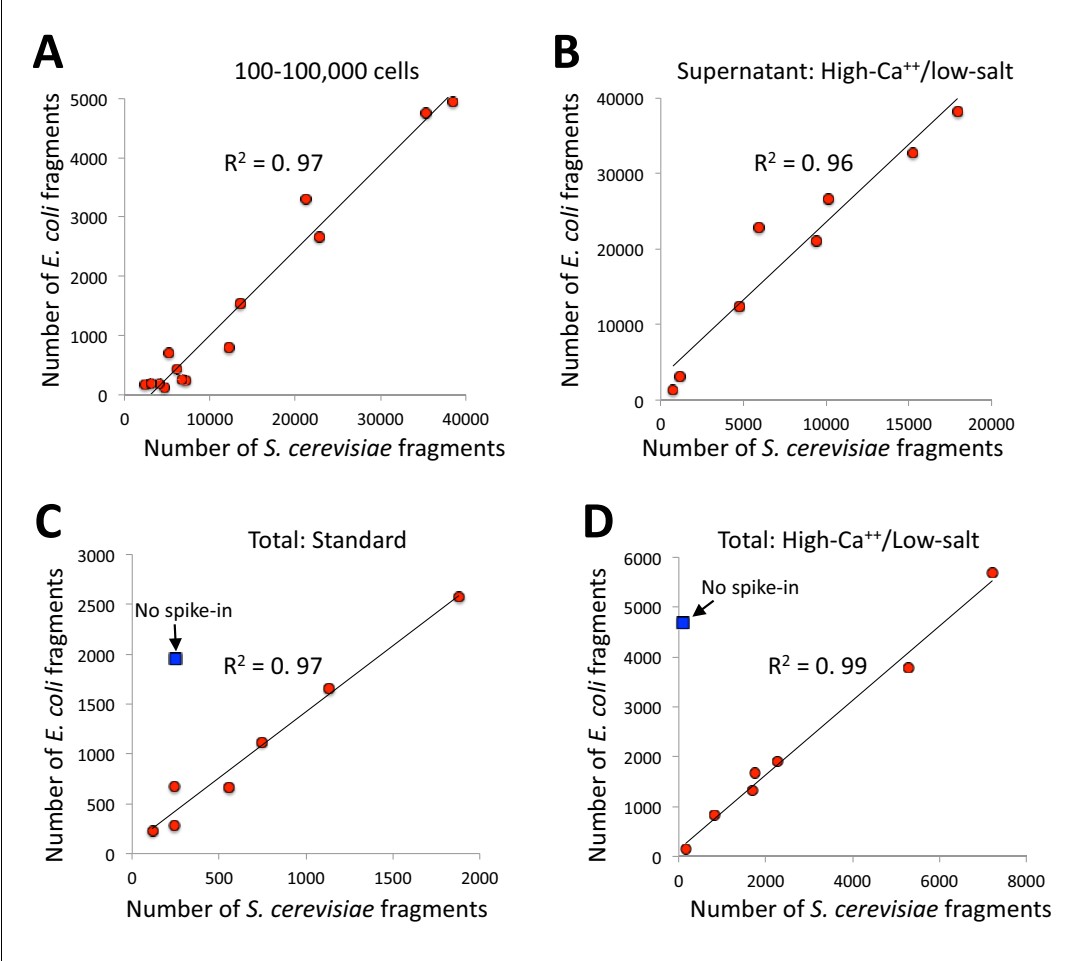

**Figure 5.** *E. coli* carry-over DNA of pA/MNase and pAG/MNase can substitute for spike-in calibration. (**A**) Fragments from a CUT&RUN K562 cell experiment (GSE104550 20170426) using antibodies against H3K27me3 (100–8,000 cells) and CTCF (1,000–100,000 cells) were mapped to the repeat-masked genome of *S. cerevisae* and the full genome of *E. coli*. Standard digestion was followed by supernatant release and extraction. (**B**) Same as A using antibodies against multiple epitopes of varying abundances, with high-calcium/low-salt digestion followed by supernatant release and extraction. (**C**) Same as B except using standard digestion conditions and total DNA extraction. The *S. cerevisiae* spike-in DNA was left out for one sample (blue square). From top to bottom, antibodies are: NPAT Thermo PA5-66839, Myc: CST Rabbit Mab #13987, CTD: PolII CTD Abcam 8WG16, RNAPII-Ser5: Abcam 5408 (mouse), RNAPII-Ser2: CST E1Z3G, CTCF Millipore 07–729, RNAPII-Ser5: CST D9N5I (rabbit), H3K4me2: Upstate 07–030. (**D**) Same as C except using high-calcium/low-salt digestion and total DNA extraction. From top to bottom, antibodies are: CTCF Millipore 07–729, NPAT Thermo PA5-66839, Myc: CST Rabbit Mab #13987, CTD: PolII CTD Abcam 8WG16, RNAPII-Ser5: Abcam 5408 (mouse), RNAPII-Ser5: CST D9N5I (rabbit), RNAPII-Ser2: CST E1Z3G, H3K4me2: Upstate 07–030.

DOI: https://doi.org/10.7554/eLife.46314.010

## Conclusions

Since its introduction in our original *eLife* paper (*Skene and Henikoff, 2017*), the advantages of CUT&RUN over ChIP-seq has led to its rapid adoption, including publication of new CUT&RUN protocols for low cell numbers (*Hainer and Fazzio, 2019*; *Skene et al., 2018*), for plant tissues (*Zheng and Gehring, 2019*) and for high-throughput (*Janssens et al., 2018*). The new CUT&RUN advances that we describe here are likely to be useful when applied in all of these protocols. Our improved CUT&RUN fusion construct simplifies reagent purification and eliminates the requirement for a secondary antibody against mouse primary antibodies. Our high-calcium/low-salt protocol minimizes time-dependent variability. Our discovery that carry-over *E. coli* DNA almost perfectly correlates with an added spike-in upgrades a contaminant to a resource that can be used as a spike-in calibration proxy, even *post-hoc* simply by counting reads mapping to the *E. coli* genome in existing CUT&RUN datasets.

**Table 1.** Carry-over *E. coli* DNA correlates closely with the heterologous spike-in for both fusion proteins and both low-salt/high-calcium and standard digestion conditions.

CUT&RUN was performed for H3K27me3 in parallel for pA/MNase Batch #6 (pA), pAG/MNase (pAG) using both low-salt/high-calcium (lo-hi) and standard (std) CUT&RUN digestion conditions. Each sample started with ~700,000 cells and 10 μL of bead slurry. Also varied in this experiment was addition of antibody followed by bead addition (Ab first) and addition of 0.1% BSA in the antibody buffer (BSA). Adding antibody first led to increased recovery of both yeast and *E. coli* DNA relative to human DNA, indicative of loss of cells prior to addition of fusion protein, possibly caused by loss of digitonin solubilization of membrane sugars.

| H3K27me3 | Ab first | BSA | Human | Yeast | *E. coli* | Corr (Sc:Ec) |
|----------|----------|-----|-------|-------|-----------|--------------|
| pA lo-hi | | | 5913983 | 743 | 3455 | 0.92 |
| pA lo-hi | | + | 7748003 | 858 | 4988 | |
| pA lo-hi | + | | 5202278 | 2288 | 16110 | |
| pA lo-hi | + | + | 5178086 | 1804 | 18759 | |
| pA std | | | 6013347 | 595 | 2462 | 0.99 |
| pA std | | + | 6005080 | 859 | 2295 | |
| pA std | + | | 4104736 | 2624 | 21236 | |
| pA std | + | + | 3972820 | 2328 | 19245 | |
| pAG lo-hi | | | 6999802 | 789 | 404 | 0.94 |
| pAG lo-hi | | + | 6374939 | 642 | 467 | |
| pAG lo-hi | + | | 4140407 | 1565 | 1291 | |
| pAG lo-hi | + | + | 4058693 | 2382 | 5289 | |
| pAG std | | | 7514127 | 308 | 567 | 0.90 |
| pAG std | | + | 5935592 | 355 | 125 | |
| pAG std | + | | 4594153 | 1271 | 555 | |
| pAG std | + | + | 5379610 | 2509 | 1353 | |

DOI: https://doi.org/10.7554/eLife.46314.011

# Materials and methods

## Key resources table

| Reagent type (species) or resource | Designation | Source or reference | Identifiers | Additional information |
|------------------------------------|-------------|---------------------|-------------|------------------------|
| Cell line (Human) | K562 | ATCC | #CCL-243 | RRID: CVCL_0004 |
| Biological sample (*Escherichia coli*) | JM101 cells | Agilent | #200234 | |
| Antibody | rabbit polyclonal anti-NPAT | Thermo | PA5-66839 | Concentration: 1:100; RRID:AB_2663287 |
| Antibody | guinea pig polyclonal anti-rabbit IgG | Antibodies Online | ABIN101961 | Concentration: 1:100; RRID: AB_10775589 |
| Antibody | rabbit polyclonal anti-mouse IgG | Abcam | 46540 | Concentration: 1:100; RRID: AB_2614925 |
| Antibody | rabbit monoclonal anti-RNAPII-Ser5 | Cell Signaling | D9N51 | Concentration: 1:100 |
| Antibody | mouse monoclonal anti-RNAPII-Ser5 | Abcam | 5408 | Concentration: 1:100; RRID:AB_304868 |
| Antibody | rabbit monoclonal anti-H3K27me3 | Cell Signaling | 9733 | Concentration: 1:100; RRID: AB_2616029 |
| Antibody | rabbit polyclonal anti-H3K4me2 | Upstate | 07–730 | Concentration: 1:100; RRID: AB_11213050 |

*Continued on next page*

*Continued*

| Reagent type (species) or resource | Designation | Source or reference | Identifiers | Additional information |
|---|---|---|---|---|
| Antibody | rabbit monoclonal anti-H3K27ac | Millipore | MABE647 | Concentration: 1:100; |
| Antibody | rabbit polyclonal anti-H3K27ac | Abcam | 4729 | Concentration: 1:100; RRID: AB_2118291 |
| Antibody | rabbit polyclonal anti-CTCF | Millipore | 07–729 | Concentration: 1:100; RRID: AB_441965 |
| Recombinant DNA reagent | AG-ERH-MNase-6xHIS-HA (plasmid) | | | Progenitors: pK19-pA-MN; gBlocks |
| Recombinant DNA reagent | pK19-pA-MN | *Schmid et al., 2004* | | Gift from author |
| Sequence-based reagent | gBlock Hemagglutinin and 6-histidine tags; gattaca GAAGACAACGCTGATTCAG GTCAAGGCGGtGGTGGcTC TGGgGGcGGgGGcTCGGGtG GtGGgGGcTCAcaccatcaccatca ccatGGCGGtGGTGGcTCTT ACCCATACGATGTTCCAGA TTACGCTtaatgaGGATCCgattaca | Integrated DNA Technologies (IDT) | | |
| Sequence-based reagent | gBLOCK PrtG_ERH Codon optimized; AGCAGAAGCTAAAAAGCTAAACGA TGCTCAAGCACCAAAAACAACTTAT AAATTAGTCATCAACGGGAAAACGC TGAAGGGTGAAACCACGACAGAGG CCGTAGATGCGGAGACAGCGGAGC GCCACTTTAAGCAATACGCGAATG ATAACGGTGTAGACGGCGAGTGGA CCTACGACGACGCGACAAAGACCT TTACCGTCACGGAGAAACCTGAGG TTATCGACGCGTCTGAGTTGACGC CAGCCGTAGATGACGATAAAGAAT TCGCAACTTCAACTAAAAAATTAC | Integrated DNA Technologies (IDT) | | |
| Peptide, recombinant protein | pA/MNase | *Schmid et al., 2004* | | purified as described in *Schmid et al., 2004* and supplementary |
| Peptide, recombinant protein | pAG/MNase | This paper | | Purified from modified plasmid pAG-ERH-MNase-6xHIS-HA in S Henikoff Lab |
| Commercial assay or kit | Pull-Down PolyHis Protein:Protein Interaction Kit | Thermo | #21277 | |
| Other | Concanavalin A coated magnetic beads | Bangs Laboratories | #BP-531 | |
| Other | Gibson Assembly | New England Biolabs | #E2611 | |
| Other | Chicken egg white lysozyme | EMD Millipore | #71412 | |
| Other | Zwittergent 3–10 detergent (0.03%) | EMD Millipore | #693021 | |
| Chemical compound, drug | Digitonin | EMD Millipore | #300410 | |
| Chemical compound, drug | Roche Complete Protease Inhibitor EDTA-free tablets | Sigma Aldrich | 5056489001 | |

*Continued on next page*

*Continued*

| Reagent type (species) or resource | Designation | Source or reference | Identifiers | Additional information |
|---|---|---|---|---|
| Chemical compound, drug | RNase A Dnase- and protease-free | Thermo | ENO531 | 10 mg/ml |
| Chemical compound, drug | Proteinase K | Thermo | EO0492 | |
| Chemical compound, drug | Glycogen | Sigma-Aldrich | 10930193001 | |
| Chemical compound, drug | Spermidine | Sigma-Aldrich | #S0266 | |

## Cell culture

K562 cells were purchased from ATCC (#CCL-243) and cultured as previously described (*Janssens et al., 2018*). All tested negative for mycoplasma contamination using MycoProbe kit.

## Construction and purification of an improved IgG-affinity/MNase fusion protein

Hemagglutinin and 6-histidine tags were added to the carboxyl-terminus of pA-MNase (*Schmid et al., 2004*) using a commercially synthesized dsDNA fragment (gBlock) from Integrated DNA Technologies (IDT), which contains the coding sequence for both tags, glycine-rich flexible linkers and includes restriction sites for cloning. Another IDT gBlock containing the optimized protein-G coding sequence and homologous flanking regions to the site of insertion, was introduced via PCR overlap extension using Gibson Assembly Master Mix (New England Biolabs cat. #E2611), following the manufacturer's instructions. The sequence-verified construct was transformed into JM101 cells (Agilent Technologies cat. #200234) for expression, cultured in NZCYM-Kanamycin (50 µg/ml) and induced with 2 mM Isopropyl β-D-1-thiogalactopyranoside following standard protein expression and purification protocols. The cell pellet was resuspended in 10 ml Lysis Buffer, consisting of 10 mM Tris-HCl pH 7.5, 300 mM NaCl, 10 mM Imidazole, 5 mM beta-mercaptoethanol, and EDTA-free protease inhibitor tablets at the recommended concentration (Sigma-Aldrich cat. #5056489001). Lysis using chicken egg white lysozyme (10 mg/mL solution, EMD cat. #71412 solution) was followed by sonication with a Branson Sonifier blunt-end adapter at output level 4, 45 s intervals for 5–10 rounds or until turbidity was reduced. The lysate was cleared by high-speed centrifugation and purified over a nickel-agarose column, taking advantage of the poly-histidine tag for efficient purification via immobilized metal affinity chromatography. Cleared lysate was applied to a 20 ml disposable gravity-flow column 1.5 ml (0.75 ml bed volume) of NI-NTA agarose (Qiagen cat. #30210), washed twice in three bed volumes of Lysis Buffer. Lysate was applied followed by two washes at five bed volumes of 10 mM Tris-HCl pH 7.5, 300 mM NaCl, 20 mM Imidazole, 0.03% ZWITTERGENT 3–10 Detergent (EMD Millipore cat. #693021) and EDTA-free protease inhibitor tablets. Elution was performed with 1 ml 10 mM Tris-HCl pH 7.5, 300 mM NaCl, 250 mM Imidazole and EDTA-free protease inhibitor tablets. Eluate was dialyzed twice against a 750 ml volume of 10 mM Tris-HCl pH 7.5, 150 mM NaCl, 1 mM EDTA, 1 mM PMSF to remove imidazole. Glycerol was then added to 50%, aliquots stored at –80°C for long term storage and –20°C for working stocks.

For purification, we used either the nickel-based protocol or the Pierce Cobalt kit (Pull-Down PolyHis Protein:Protein Interaction Kit cat. #21277 from Thermo Fisher). Similar results were obtained using either the nickel or cobalt protocol, although the cobalt kit alleviated the need for a sonicator, using a fifth of the starting material from either fresh culture or a cell pellet frozen in lysis buffer, and yielded more protein per volume of starting material. With the cobalt kit, 20 ml of culture yielded ~100 µg of fusion protein.

## CUT&RUN using high-calcium/low-salt digestion conditions

Log-phase cultures of K562 cells were harvested, washed, and bound to activated Concanavalin A-coated magnetic beads, then permeabilized with Wash buffer (20 mM HEPES, pH7.5, 150 mM NaCl, 0.5 mM spermidine and a Roche complete tablet per 50 ml) containing 0.05% Digitonin (Dig-Wash) as described (*Skene et al., 2018*). The bead-cell slurry was incubated with antibody in a 50–

100 µL volume for 2 hr at room temperature or at 4°C overnight on a nutator or rotator essentially as described (*Skene et al., 2018*). In some experiments, cells were permeabilized and antibody was added and incubated 2 hr to 3 days prior to addition of ConA beads with gentle vortexing; similar results were obtained (*e.g. Figure 2B–D*), although with lower yields. After 2–3 washes in 1 ml Dig-wash, beads were resuspended in 50–100 µL pA/MNase or pAG/MNase and incubated for 1 hr at room temperature. After two washes in Dig-wash, beads were resuspended in low-salt rinse buffer (20 mM HEPES, pH7.5, 0.5 mM spermidine, a Roche mini-complete tablet per 10 ml and 0.05% Digitonin). Tubes were chilled to 0°C, the liquid was removed on a magnet stand, and ice-cold calcium incubation buffer (3.5 mM HEPES pH 7.5, 10 mM $CaCl_2$, 0.05% Digitonin) was added while gently vortexing. Tubes were replaced on ice during the incubation for times indicated in each experiment, and within 30 s of the end of the incubation period the tubes were replaced on the magnet, and upon clearing, the liquid was removed, followed by immediate addition of EGTA-STOP buffer (170 mM NaCl, 20 mM EGTA, 0.05% Digitonin, 20 µg/ml glycogen, 25 µg/ml RNase A, 2 pg/ml *S. cerevisiae* fragmented nucleosomal DNA). Beads were incubated at 37°C for 30 min, replaced on a magnet stand and the liquid was removed to a fresh tube and DNA was extracted as described (*Skene et al., 2018*). A detailed step-by-step protocol is available at https://www.protocols.io/view/cut-amp-run-targeted-in-situ-genome-wide-profiling-zcpf2vn. Extraction of pellet and total DNA was performed essentially as described (*Skene and Henikoff, 2017*; *Thakur and Henikoff, 2018*).

## DNA sequencing and data processing

The size distribution of libraries was determined by Agilent 4200 TapeStation analysis, and libraries were mixed to achieve equal representation as desired aiming for a final concentration as recommended by the manufacturer. Paired-end Illumina sequencing was performed on the barcoded libraries following the manufacturer's instructions. Paired-end reads were aligned using Bowtie2 version 2.2.5 with options: `--local --very-sensitive-local --no-unal --no-mixed --no-discordant --phred33 -I 10 -X 700`. For MACS2 peak calling, parameters used were `macs2 callpeak – t input_file –p 1e-5 –f BEDPE/BED(Paired End vs. Single End sequencing data) –keep-dup all –n out_name`. Some datasets showed contamination by sequences of undetermined origin consisting of the sequence $(TA)_n$. To avoid cross-mapping, we searched blastn for TATATATATATATATATATATATAT against hg19, collapsed the overlapping hits into 34,832 regions and intersected with sequencing datasets, keeping only the fragments that did not overlap any of these regions.

## Evaluating time-course data

If digestion and fragment release into the supernatant occur linearly with time of digestion until all fragments within a population are released, then we expect that CUT&RUN features will be linearly correlated within a time-course series. For CTCF, features were significant CTCF motifs intersecting with DNAseI hypersensitive sites (*Skene and Henikoff, 2017*). For H3K27Ac and H3K4me2, we called peaks using MACS2 and calculated the Pearson correlation coefficients between time points, displayed as a matrix of $R^2$ values, using the following procedure:

1. Aligned fastq files to unmasked genomic sequence using Bowtie2 version 2.2.5 to UCSC hg19 with parameters: `--end-to-end --very-sensitive --no-mixed --no-discordant -q --phred33 -I 10 -X 700`.
2. Extracted properly paired read fragments from the alignments and pooled fragments from multiple samples.
3. Compared pooled fragments with $(TA)_n$ regions of hg19 and kept those fragments that did NOT overlap any $(TA)_n$ region using bedtools 2.21.0 with parameters: `intersect -v -a fragments.bed -b TATA_regions.bed>fragments_not_TATA.bed`.
4. Found peaks using macs2 2.1.1.20160309 with parameters: `callpeak -t fragments_not_-mask.bed -f BED -g hs –keep-dup all -p 1e-5 -n not_mask –SPMR`.
5. Made scaled fractional count bedgraph files for each sample from bed files made in step 2. The value at each base pair is the fraction of counts times the size of hg19 so if the counts were uniformly distributed the value would be one at each bp.
6. Extracted bedgraph values for ±150 bps around peak summits for IgG sample and computed their means, which resulted in one mean score per peak.

7. Removed peaks from macs2 results in step four if the mean score was greater than the 99th percentile of all IgG scores to make a subset of the peaks lacking the most extreme outliers.
8. Extracted bedgraph values for ±150 bps around the subset of peak summits from step seven for all samples and computed their means, which resulted in a matrix with columns corresponding to samples and one row per peak.
9. Computed correlations of matrix in eight using R 3.2.2 `cor(matrix, use='complete.obs')` command.

## Acknowledgements

We thank Christine Codomo and Tayler Hentges for technical support. We also thank all members of the Henikoff lab for valuable discussions and Kami Ahmad, Brian Freie and Bob Eisenman for comments on the manuscript. This work was supported by the Howard Hughes Medical Institute, and a grant from the National Institutes of Health (4DN TCPA A093) and the Chan-Zuckerberg Initiative.

## Additional information

### Funding

| Funder | Grant reference number | Author |
| --- | --- | --- |
| Howard Hughes Medical Institute | | Steven Henikoff |
| National Institutes of Health | 4DN TCPA A093 | Steven Henikoff |
| Chan-Zuckerberg Initiative | | Steven Henikoff |

The funders had no role in study design, data collection and interpretation, or the decision to submit the work for publication.

### Author contributions

Michael P Meers, Validation, Writing—review and editing; Terri D Bryson, Investigation, Methodology, Writing—original draft, Writing—review and editing; Jorja G Henikoff, Data curation, Software, Formal analysis, Writing—review and editing; Steven Henikoff, Conceptualization, Resources, Supervision, Funding acquisition, Validation, Investigation, Visualization, Methodology, Writing—original draft, Writing—review and editing

### Author ORCIDs

Steven Henikoff https://orcid.org/0000-0002-7621-8685

### Decision letter and Author response

Decision letter https://doi.org/10.7554/eLife.46314.017
Author response https://doi.org/10.7554/eLife.46314.018

## Additional files

### Supplementary files

• Transparent reporting form
DOI: https://doi.org/10.7554/eLife.46314.012

### Data availability

The plasmid pAG-ERH-MNase-6xHIS-HA is available from Addgene. Sequencing datasets are available from GEO (GSE126612).

The following dataset was generated:

| Author(s) | Year | Dataset title | Dataset URL | Database and Identifier |
|---|---|---|---|---|
| Meers MP, Bryson TD, Henikoff S | 2019 | A streamlined protocol and analysis pipeline for CUT&RUN chromatin profiling | http://www.ncbi.nlm.nih.gov/geo/query/acc.cgi?acc=GSE126612 | NCBI Gene Expression Omnibus, GSE126612 |

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
