## [Decision Letter]

Thank you for submitting your article "A streamlined protocol and analysis pipeline for CUT&RUN chromatin profiling" for consideration by *eLife*. Your article has been reviewed by three peer reviewers, and the evaluation has been overseen by a Reviewing Editor and Detlef Weigel as the Senior Editor. The reviewers have opted to remain anonymous.

The reviewers have discussed the reviews with one another and the Reviewing Editor has drafted this decision to help you prepare a revised submission.

Summary:

The main objective of the manuscript is to introduce an improved protocol for CUT&RUN and a peak calling algorithm. The authors made optimized the pA-MNase enzyme (now pAG-MNase), for easier purification and recognition of both mouse and rabbit primary antibodies. Furthermore, the authors suggest an improved high Ca^2+^/low salt CUT&RUN protocol that prevents overdigestion/nonspecific digestion. The authors also find that *E. coli* DNA carried over in the pAG-MNase purification is still present in CUT&RUN sequencing samples and can therefore be used to normalize CUT&RUN data. Lastly, a new peak calling algorithm is proposed for calling peaks in CUT&RUN data as it typically has low read number and high signal to noise ratio. Although this manuscript does not contain any biological findings or major changes to the current CUT&RUN protocol, it does communicate important improvements to a technique that many labs are interested in using.

Essential revisions:

1) All reviewers uniformly shared major concerns about the peak calling algorithm, and we summarize these here:

a) There are several different modes and the description how they differ and when each is appropriate wasn't clear.

b) The sudden drop-off of SEACR with more data (going from 25M to 30M reads, Figure 5D) reveals a very concerning flaw in the model or a bug in the implementation. Performance should improve with more data in a downsampling experiment. The overall drop-off and poor performance present significant questions to its claim of robustness and general usability.

c) Could MACS2 and HOMER algorithms perform comparably to SEACR simply by tuning their parameters (wasn't clear how much of an effort was made to do this, and we think that is important for algorithm benchmarking)? More comparisons to other factors would be helpful.

d) SEACR does not provide any estimate of statistical significance to the assigned peaks compared to other methods. How are users to interpret confidence in peak calls?

e) Why were the target blocks defined by contiguous regions of nonzero coverage rather than tiling windows?

f) Figure 5 – label axes of (A)-(C) more clearly. These appear to be TPR vs. FPR using the encode logFDR<-10 peaks as a truth set; is that right? But should each peak caller at a given sampling depth have a summary value (auROC or auPR) rather than a point?

g) Figure 6B – SEACR is more aggressive in aggregating long peaks but this seems sort of trivial (i.e. one could do something similar by padding and merging peaks)

Overall, we felt that the other aspects of the manuscript were strong enough to warrant a path forward in the Research Advances format even if all the above items about the peak caller are not addressable.

2) A high Calcium/Low salt procedure is included that reduces diffusion of the released complex. The data to support this lower diffusion is that signal to noise appears higher. However, direct evidence for lower diffusion within the nucleus is not provided. Background seems to be lower in specific example loci (e.g. shown in Figure 1B) – this should be quantified genome-wide (e.g.,% signal in peaks). The improvement appears to be more profound in some examples (H3K27ac, Figure 2—figure supplement 1A) than others (H34Kme2, Figure 2—figure supplement 1A). Also, are inter-sample correlations the best way to show signal:noise improvement? It would be more convincing with precision and recall (or similar) vs. high-confidence peaks.

3) The authors claim that DNA carry over from *E. coli* in the pAG-MN preparation is a good substitute for the yeast genomic DNA spike in that is normally used. My concern about this is that this is not a well-controlled spike in, as the amount of *E. coli* DNA may well vary between batches of the purified PAG-MNase.

[Editors' note: further revisions were requested prior to acceptance, as described below.]

Thank you for resubmitting your work entitled "Improved CUT&RUN chromatin profiling and analysis tool" for further consideration at *eLife*. Your revised article has been favorably evaluated by Detlef Weigel (Senior Editor), a Reviewing Editor, and three reviewers.

The manuscript has been improved but there are some remaining issues that need to be addressed before acceptance, as outlined below:

All three reviewers have examined the revised manuscript and feel that all points were addressed with the exception of the issues raised in items 1a-g regarding the peak caller. Importantly, all reviewers remain uniformly concerned about this aspect of the work and do not believe the revisions adequately addressed the points that were outlined. All reviewers believe that other aspects of this manuscript warrant publication and advocate for moving forward without the peak caller, which would expedite publication.

However, if the authors would like to make additional revisions to the peak caller sections, we will re-review those and recommend directly addressing the items we initially raised:

1b) The response to this point was to add an ad hoc "genome coverage" filter to effectively throw out reads and decrease noise. This is not a reasonable solution for a robust peak caller and only emphasizes the sensitivity of the method to read depth and noise.

1c) Insufficient parameter exploration was provided to convincingly demonstrate that the other peak callers are inferior to SEACR. For example, in addition to changing MACS2 FDR, why have the authors not attempted to change the local λ smoothing parameter, which would make MACS2 use a genome-wide Poisson threshold that is more equivalent to the single genome-wide threshold that SEACR uses? More informed parameter exploration beyond the single example provided here would make a stronger case.

1d) The authors state that statistical model based FDRs are inferior to their empirical threshold, but the performance results only support such a statement under a limited and author-selected (yellow highlights in plots) range of read depths. Further, performance on additional antibodies (TFs, histone mods, etc.) should be presented before such a strong claim is made.

1e) If "contiguous signal blocks reflect real patterns of protein protection that should be incorporated into the peak calls." then why do the authors need to implement a "genome coverage" filter at high read depths? This statement is contradictory to the performance results and methods in the manuscript.

1f) See point 1c above.

Overall, the SEACR algorithm seems very sensitive to background noise levels, which may be highly variable across diverse labs that will implement the revised CUT&RUN technique. This could be a recipe for confusion and misinterpretation across the user base. For the collective reasons outlined above, we remain skeptical about including the peak caller in this manuscript. However, we would welcome immediate forward movement if this section is removed.

---

## [Author Response]

Summary:The main objective of the manuscript is to introduce an improved protocol for CUT&RUN and a peak calling algorithm. The authors made optimized the pA-MNase enzyme (now pAG-MNase), for easier purification and recognition of both mouse and rabbit primary antibodies. Furthermore, the authors suggest an improved high Ca^2+^/low salt CUT&RUN protocol that prevents overdigestion/nonspecific digestion. The authors also find that E. coli DNA carried over in the pAG-MNase purification is still present in CUT&RUN sequencing samples and can therefore be used to normalize CUT&RUN data. Lastly, a new peak calling algorithm is proposed for calling peaks in CUT&RUN data as it typically has low read number and high signal to noise ratio. Although this manuscript does not contain any biological findings or major changes to the current CUT&RUN protocol, it does communicate important improvements to a technique that many labs are interested in using.

We thank the reviewers and editors for their appreciation of the importance of our improvements to the CUT&RUN method and for their very helpful comments and suggestions, which we address below.

Essential revisions:1) All reviewers uniformly shared major concerns about the peak calling algorithm, and we summarize these here: a) There are several different modes and the description how they differ and when each is appropriate wasn't clear.

We agree that the utility of the different modes was not made explicitly clear in the text, especially between the “AUC” and the “union” modes. As described when testing AUC and union mode on Sox2 and FoxA2 data, union mode was introduced in order to improve the recall of SEACR for narrow, tall peaks that do not meet the total signal threshold assigned in AUC mode. Therefore, AUC mode defines a highly stringent set of peaks, whereas union mode generates more peaks with a small precision penalty. To clarify the distinction, we have changed the names of the modes from “AUC” and “union” to “stringent” and “relaxed” and have more extensively described their differences in the text (subsection “Peak-calling based on fragment block aggregation”). Moreover, we are in the process of implementing a SEACR web server to facilitate its broad use by non-technical users, and we intend to report outputs for both modes so that users can get a sense of the precision-recall tradeoffs for each, and use their preferred mode accordingly.

b) The sudden drop-off of SEACR with more data (going from 25M to 30M reads, Figure 5D) reveals a very concerning flaw in the model or a bug in the implementation. Performance should improve with more data in a downsampling experiment. The overall drop-off and poor performance present significant questions to its claim of robustness and general usability.

With respect to the possibility that the “model” is flawed, it should be understood that SEACR does not use a model at all—the total signal threshold is derived from empirical distributions of signal, which is the source of its simplicity and relative speed (SEACR with input files representing ~10M fragments typically completes in less than two minutes). SEACR was designed to deal with CUT&RUN data that is sequenced to relatively low read depths as compared to ChIP-seq (hence Sparse Enrichment Analysis). Our original *eLife* manuscript demonstrated that CUT&RUN can generate signal-to-noise that is far superior to ChIP-seq with fewer than 10 million fragments from a human experiment, and in the examples in the current manuscript we sequence only 3-5 million fragments, because deeper sequencing provides little if any gain. SEACR’s performance declines once the sequence depth is high enough that background signal accumulates, and this is responsible for the relatively poor SEACR performance above 30 million fragments in Figure 5D (new Figure 7D).

To address this, we have implemented a “genome coverage” filter that requires that greater than 50% of the reference genome lacks read coverage, therefore ensuring that the distribution of background signal is sparse enough for SEACR. We accomplish this by finding a minimum bedgraph signal threshold *n* for which converting regions of less than *n* signal to 0 results in satisfying the aforementioned 50% threshold. In the new Figure 7—figure supplement 1A, we show that when we calculate F1 scores for this amended implementation of SEACR similarly to Figure 5D (new Figure 7D), SEACR outperforms MACS2 and HOMER at 30M or more fragments, mirroring its better performance at lower fragment depths.This is a general solution, since no transcription factor or chromatin feature is expected to span this much of the genome. We should emphasize that SEACR is effective at avoiding false negatives in both transcription factor and histone modification CUT&RUN data while providing sufficient recall to outperform MACS2 and HOMER at low sequencing depth. We believe that retaining the core code and presenting its flaws in the extreme cases serves as a warning not to waste money by unnecessary sequencing.

c) Could MACS2 and HOMER algorithms perform comparably to SEACR simply by tuning their parameters (wasn't clear how much of an effort was made to do this, and we think that is important for algorithm benchmarking)? More comparisons to other factors would be helpful.

It is possible to improve the performance of MACS2 and HOMER by tuning parameters, and we have added analysis and discussion of this to the manuscript (new Figure 7—figure supplement 1B). However, the point of SEACR is that it uses a completely different design for calling peaks, and this algorithm works best at low sequencing depths. In comparing MACS2 and HOMER to SEACR, we made sure for each to use the “mode” that was appropriate for the data being analyzed according to the user guides for each algorithm. For example, we used MACS2 “narrow peak” mode and HOMER “factor” mode to call peaks from H3K4me2 data, and used “broad” and “histone” mode to analyze H3K27me3 data. This is newly emphasized in the text (subsection “Peak-calling based on fragment block aggregation”).

To the extent that parameters such as MACS2 FDR can be tuned to modulate the precision-recall balance, this burdens users with having to make arbitrary decisions about their peak calls, while SEACR is designed to provide a single empirical threshold for the user. Nevertheless, in this revision we provide an example of how selecting specific parameters for MACS2 changes its performance relative to SEACR in new Figure 7—figure supplement 1B. We took the MACS2 peak calls for H3K4me2 that we originally presented, and selected only peaks that met a minimum –log_10_(FDR) threshold of 10, in order to tilt the precision-recall balance in favor of precision similar to SEACR. In doing so and calculating F1 scores as previously described, we found that although the new MACS2 peak calls performed similarly to SEACR across the upper end of the optimal range of fragment depths that we originally defined in Figure 5D (new Figure 7D), its performance suffered dramatically at low fragment depths, making SEACR newly superior at those subsampling levels. Therefore, we conclude that, even when purposefully selecting parameters for a different peak caller to tune its performance, SEACR performs competitively in the absence of any arbitrary user input.

d) SEACR does not provide any estimate of statistical significance to the assigned peaks compared to other methods. How are users to interpret confidence in peak calls?

Since SEACR is model-free, we cannot assign statistics such as false discovery rate to each peak based on a comparison to a statistical model, and we consider model statistics inferior to the threshold we derive from actual empirical data. However, SEACR already calculates the threshold at which the fraction of remaining signal blocks from the target dataset is maximized by finding the following value:

Max(1-(IgG blocks/total blocks)) = 1-Min(IgG blocks/total blocks)

Since the IgG blocks/total blocks term is analogous to an empirically calculated false discovery rate (FDR), we now report the minimum value for this term in the standard output of SEACR in order to convey the confidence inherent to the global threshold. Users can interpret this value as they would a standard FDR and use it to determine the general quality of the peak calls as a whole.

Although we can report an FDR, we are uncomfortable with the inclusion of confidence statistics in peak-caller output, because this risks inspiring false confidence on the part of a non-technical user. Therefore, in our planned public web server we will make it an option with the following warning: “Assigning confidence measures for individual peaks often falsely gives the impression that the peak is in fact a true positive, in the absence of a “gold standard” to verify this impression.” Our demonstration that SEACR calls ~10,000 true positives (Sox2 in ESCs and FoxA2 in Endoderm where they are expressed) but only 1-2 false positives (Sox2 in Endoderm and FoxA2 in ESCs where they are not expressed) achieves for CUT&RUN gold-standard validation. Such validation has never been achieved for ChIP-seq, where for example reports of “Phantom Peaks” in modENCODE data imply that upwards of 25% of peaks are called regardless of the presence or absence of the targeted factor (Jain et al., 2015). We now emphasize this point in the text (subsection “Conclusions”).

e) Why were the target blocks defined by contiguous regions of nonzero coverage rather than tiling windows?

CUT&RUN digestion patterns are informative in a way that ChIP-seq random fragmentation is not, and therefore contiguous signal blocks reflect real patterns of protein protection that should be incorporated into the peak calls. There are additional benefits. CUT&RUN datasets typically have very sparse background signal, meaning we can take advantage of this characteristic by understanding that contiguous signal blocks with the highest total signal contained within them are most likely to be true positive peaks, rendering tiling unnecessary. From a design perspective, we were interested in using the empirical data rather than abstracting from it. From a practical perspective, window tiling is much more computationally intensive than our data-driven approach. For example, window tiling requires that one discard data unless the tiling is done per-base, which incurs a large computational cost, and therefore we would sacrifice speed where it isn’t clear that window tiling would even be a preferred approach.

f) Figure 5 – label axes of (A)-(C) more clearly. These appear to be TPR vs. FPR using the encode logFDR<-10 peaks as a truth set; is that right? But should each peak caller at a given sampling depth have a summary value (auROC or auPR) rather than a point?

The reviewers are correct that ENCODE peaks of -log_10_(FDR) > 10 is used as the “truth set”; we make this more explicit in the text (subsection “Peak-calling based on fragment block aggregation”) and have made the axis labels more clear. If we understand the final question correctly, the reviewers are proposing that some parameter of the peak calls (e.g. FDR from MACS2) be varied such that a curve can be derived, rather than a single point from the full peak call set. As was outlined above in response to point (c), we feel this would unfairly disadvantage SEACR since it is designed to find a single threshold that reflects high precision in peak calling. However, we also point the reviewers to the new Figure 7—figure supplement 1B produced in response to point (c), in which MACS2 FDR is varied and F1 scores calculated, which partially addresses this proposal by showing that SEACR remains competitive in comparison to MACS2 across multiple parameter selection strategies for MACS2.

g) Figure 6B – SEACR is more aggressive in aggregating long peaks but this seems sort of trivial (i.e. one could do something similar by padding and merging peaks)

As is the case with other points raised above, parameters could be selected from other peak callers or other manual means used to achieve a desired degree of “peak-stitching”. However, the ability to generate domains without arbitrary user input remains valuable. Moreover, SEACR does not require manual tuning for data with vastly different patterns (e.g. H3K4me2 and H3K27me3).

Overall, we felt that the other aspects of the manuscript were strong enough to warrant a path forward in the Research Advances format even if all the above items about the peak caller are not addressable.

Our revisions to the implementation and description of SEACR should address the reviewers’ concerns.

2) A high Calcium/Low salt procedure is included that reduces diffusion of the released complex. The data to support this lower diffusion is that signal to noise appears higher. However, direct evidence for lower diffusion within the nucleus is not provided.

The direct evidence for reduced diffusion is presented in Figure 2, where we show that incubation at 37^o^C for 30 minutes results in extraction of the nucleosome ladder for H3K27me3 and H3K27ac under standard conditions, but no detectable release under high-salt/low calcium conditions. The 10 mM Ca^++^ 3.5 mM HEPES pH7.5 condition that we used was based on the observation that mononucleosomes that are freely soluble without divalent cations are insoluble in 10 mM Ca^++^ or Mg^++^. To avoid any misunderstanding on the physical basis for the observation shown in Figure 2, we have removed the term “diffusion” where it was used in the text, and now describe this phenomenon consistently as “premature release”.

Background seems to be lower in specific example loci (eg shown in Figure 1B) – this should be quantified genome-wide (e.g.,% signal in peaks).

Figure 1B was intended to show that pAG/MNase works equally well for both a rabbit monoclonal and a mouse monoclonal antibody using RNAPII-Ser5P as an example, and we now show representative tracks that make this point. We show tracks for a histone gene cluster, because the genes are small and close together so that the distinction between signal and background is readily apparent. However, it is problematic to call peaks on RNAPII-Ser5P, genome-wide because this modification is enriched over 5’ ends of genes but also present throughout gene bodies at a low level. Instead, we now show identically scaled tracks for an H3K4me2 CUT&RUN experiment done in parallel with the RNAPII-Ser5P experiment shown in Figure 1B (new Figure 4—figure supplement 1B). It is apparent from the tracks that the backgrounds are extremely low for both the standard protocol and the high-calcium/low-salt protocol. As requested, we also provide a Fraction of reads in peaks (Frip) plot, the ENCODE-recommended standard for evaluating relative data quality (new Figure 4 and Figure 4—figure supplement 1C).

The improvement appears to be more profound in some examples (H3K27ac, Figure 2—figure supplement 1A) than others (H34Kme2, Figure 2—figure supplement 1A). Also, are inter-sample correlations the best way to show signal:noise improvement? It would be more convincing with precision and recall (or similar) vs high-confidence peaks.

We have added representative tracks and Frip plots for the time points, which show improved signal-to-noise (new Figure 4C). However, there seems to be a misunderstanding of what we were intending to show with the correlation matrices. Our conclusion was that the new protocol provides higher yields by preventing premature release, which can increase background with longer digestion times, and we have made textual changes in the revision to make this point clearer (subsection “Preventing premature release during CUT&RUN digestion”). The correlation matrices were meant to show that over the course of digestion there is improved uniformity and this improvement is confirmed in the Frip plots. As we first documented in our original *eLife* paper and confirmed in subsequent publications (PMID: 29651053; 30577869; 30554944), backgrounds for CUT&RUN using the standard protocol are much lower than for ChIP-seq (e. g.Figure 1—figure supplement 2). Since then, we have distributed CUT&RUN materials to >600 laboratories, and although feedback has been mostly positive, a common problem that users ask us about is relatively high background with some antibodies, which we have traced to overdigestion (we recommend 30 minutes at 0^o^C to maximize yields). We chose to do most of our testing on H3K27ac antibodies from Abcam and Millipore because these are the antibodies that we and others have had the most background problems with, perhaps because H3K27 acetylation is so abundant that release during digestion happens more rapidly than for transcription factors and most other epitopes. In fact, we see no consistent signal-to-noise differences between the standard and high-calcium/low-salt conditions for H3K4me2, now documented in the new Figure 4—figure supplement 1 described above. Because there are already hundreds of satisfied CUT&RUN users who follow the standard protocol described on Protocols.io (version 3, https://www.protocols.io/view/cut-amp-run-targeted-in-situ-genome-wide-profiling-zcpf2vn/guidelines), we are only recommending high-salt/low-calcium conditions as an option “for targets that are enriched at active chromatin (e.g. H3K27ac)”.

3) The authors claim that DNA carry over from E. coli in the pAG-MN preparation is a good substitute for the yeast genomic DNA spike in that is normally used. My concern about this is that this is not a well-controlled spike in, as the amount of E. coli DNA may well vary between batches of the purified PAG-MNase.

The goal of calibration is to allow for comparisons to be made in a series, so as long as a user follows the protocol, this issue will never arise, especially insofar as a single batch of enzyme using the Pierce kit yields enough for ~10,000 samples. Using two different lots of any critical reagent (e.g. an antibody) for samples in a series would be contrary to accepted laboratory practice. Even if a user runs short of pAG/MNase, mixing batches is OK, as long as the mixing is done before adding to samples in a series. We now state in the text that the *E. coli* carry-over DNA can be used for samples in a series using the same batch of pAG-MNase (subsection “Calibration using *E. coli* carry-over DNA”).

[Editors' note: further revisions were requested prior to acceptance, as described below.]

The manuscript has been improved but there are some remaining issues that need to be addressed before acceptance, as outlined below:All three reviewers have examined the revised manuscript and feel that all points were addressed with the exception of the issues raised in items 1a-g regarding the peak caller. Importantly, all reviewers remain uniformly concerned about this aspect of the work and do not believe the revisions adequately addressed the points that were outlined. All reviewers believe that other aspects of this manuscript warrant publication and advocate for moving forward without the peak caller, which would expedite publication.

We are glad that the reviewers found our responses to be satisfactory for most of the manuscript, and we agree that the peak-caller needs additional work. Therefore, as requested by the reviewers, we have removed the peak-caller in its entirety from the manuscript to expedite publication. This required removal of a sentence from the Abstract, a sentence from the Introduction, a sub-section from the Results section and Discussion section, a paragraph from the Conclusions, a sub-section from the Materials and methods section and Figure 6, Figure 7, Figure 6—figure supplement 1 and Figure 7—figure supplement 1. We have made minor textual changes in accordance with the removal of SEACR and have changed the title to “Improved CUT&RUN chromatin profiling tools”.

However, if the authors would like to make additional revisions to the peak caller sections, we will re-review those and recommend directly addressing the items we initially raised:

As described above, the peak caller section and associated figures and text have been removed.

Overall, the SEACR algorithm seems very sensitive to background noise levels, which may be highly variable across diverse labs that will implement the revised CUT&RUN technique. This could be a recipe for confusion and misinterpretation across the user base. For the collective reasons outlined above, we remain skeptical about including the peak caller in this manuscript. However, we would welcome immediate forward movement if this section is removed.

We agree that our manuscript without SEACR is ready to move forward, as all requirements for publication have been met.